# Stallion Sperm Freezing with Different Extenders: Role of Antioxidant Activity and Nitric Oxide Production

**DOI:** 10.3390/ani14172465

**Published:** 2024-08-25

**Authors:** Raffaele Boni, Raffaella Ruggiero, Tommaso Di Palma, Maria Antonietta Ferrara, Graziano Preziosi, Stefano Cecchini Gualandi

**Affiliations:** 1Department of Basic and Applied Sciences (DiSBA), University of Basilicata, Via dell’Ateneo Lucano, 10, 85100 Potenza, Italy; raffaella5ruggiero@gmail.com; 2Breeders’ Association (A.R.A.) of Basilicata, Via dell’Edilizia, 85100 Potenza, Italy; tao.tao.tdp@gmail.com; 3Institute of Applied Sciences and Intelligent Systems, Unit of Naples, Italian National Research Council (ISASI-CNR), Via Pietro Castellino 111, 80131 Napoli, Italy; antonella.ferrara@na.isasi.cnr.it (M.A.F.); graziano.preziosi@na.isasi.cnr.it (G.P.)

**Keywords:** equine, sperm quality, total antioxidant capacity, nitric oxide stable metabolites, freezing injuries, conditioned extenders

## Abstract

**Simple Summary:**

Freezing stallion semen yields variable results, heavily dependent on individual differences, categorizing stallions as good or poor freezers. This variability was investigated by analyzing sperm kinetics, mitochondrial membrane potential, and hydrogen peroxide content in sperm samples treated with different extenders and procedures. Additionally, antioxidant activities based on ABTS and FRAP assays, along with nitric oxide stable metabolites (NOx), were assessed in seminal plasma, blank extenders, and extenders conditioned by spermatozoa either before or after freezing. Strong individual variability was observed in most sperm functions both before and after freezing. Several physical and chemical differences were noted between the shipping and freezing extenders. However, no significant differences in sperm function, antioxidant activities, and NOx values were found among the shipping extenders. In contrast, significant differences in antioxidant activities and NOx values were found between the freezing extenders. The significant interaction between stallion and extender suggests that customizing freezing procedures can improve sperm freezing outcomes in stallions.

**Abstract:**

Sensitivity to freezing remains a critical issue in stallion semen cryopreservation procedures. To explore this topic in-depth, semen was collected from ten stallions, diluted with three different extenders, transported to the laboratory, and then centrifuged and frozen with four different extenders. We conducted analyses of sperm kinetics, mitochondrial membrane potential (MMP), and hydrogen peroxide content both before and after freezing. Additionally, we assessed antioxidant activity using the ABTS and FRAP methods and measured nitric oxide stable metabolites (NOx) in the blank extenders, seminal plasma, and extenders conditioned by spermatozoa before and after freezing. We found significant variability in the antioxidant activity and NOx content of the blank extenders and the seminal plasma. In the seminal plasma, ABTS-based antioxidant activity and NOx values were correlated with some sperm kinematic parameters and MMP in refrigerated semen, while no correlation was observed in frozen sperm parameters. Sperm function varied significantly between stallions but not between extenders, either before or after freezing. However, significant differences in antioxidant activities and NOx values were found among extenders conditioned following freezing. These results provide new insights into the factors contributing to the variability in individual stallions’ tolerance to sperm freezing.

## 1. Introduction

Freezing procedures can harm sperm survival and fertility due to the physical and chemical stress exerted on cells during freezing. This stress is related to several factors: (i) the progressive increase in the osmolarity of the extender due to the transformation of free water into ice crystals [1]; (ii) the high osmolarity of the freezing extender due to the supplementation with cryoprotectant agents (CPAs) [1,2]; (iii) the rate of intracellular diffusion of permeant CPAs and water exchange, which varies depending on CPA type and concentration, cell type, animal species, temperature, and other factors [3]; (iv) mechanical damage to the integrity of cell membranes, other cellular structures, and genetic material caused by ice crystal formation [4]; (v) structural and functional damage to lipid components, known as lipoperoxidation [5]; (vi) the toxicity of CPAs [6].

Penetrating CPAs are generally small, non-ionic molecules that can easily diffuse through cell membranes and protect biological samples from freezing injuries during cryopreservation. This protection is achieved through mechanisms that depend on the specific CPAs used, which are not yet fully understood [7]. In addition to their cryoprotectant properties, some CPAs also have antioxidant roles. For instance, glycols, which are alcohols containing at least two hydroxyl groups and are used as CPAs (such as ethylene glycol, propylene glycol, and glycerol), are not primarily known for antioxidant activity. However, glycerol, the first compound discovered to allow sperm survival at cryogenic temperatures [8], has demonstrated antioxidant activity when combined with ethanol [9]. Dimethyl sulfoxide (DMSO), the most widely used CPA for freezing cell lines [10], has powerful antioxidant activity [11]. DMSO belongs to a class of polar aprotic solvents, which includes dimethylformamide (DMF). The cryoprotectant value of DMF has been recently highlighted for addressing the issue of poor sperm freezability with glycerol [12]. DMF has also demonstrated antioxidant activities related to its interference with the hydroxyl-radical transduction pathway [13]. However, like many antioxidant substances, DMF can become pro-oxidizing and cause pathological occurrences, such as endoplasmic reticulum stress, when combined with other substances [14].

Spermatozoa are highly sensitive to oxidative stress [15]. This sensitivity may be due to the following: (i) motility dysfunctions following decreased membrane fluidity caused by lipid peroxidation [16]; (ii) direct oxidation of proteins and membrane permeabilization [17]; (iii) a lack of effective antioxidant protection systems due to a scarcity of cytoplasm [18]. These issues have inevitable repercussions on fertility. Reactive oxygen and nitrogen species (ROS and RNS, collectively known as RONS) can be found in the extracellular environment or inside the cell, mostly generated by cellular metabolism and respiration. While increased cellular activity is generally a positive indicator of sperm quality, if RONS accumulate without compensation or dispersion, as may occur in spermatozoa inside the female genital tract, they compromise sperm quality. At physiological concentrations, RONS activate intracellular pathways that underlie sperm maturation, capacitation, hyperactivation, acrosome reaction, and gamete fusion [15]. However, when ROS production overwhelms cellular antioxidant defense systems, oxidative stress occurs.

One of the primary sources of ROS production in cells is the mitochondria, whose activity can be indirectly assessed by measuring the mitochondrial membrane potential (MMP)—a parameter positively correlated with sperm function, particularly in mammals [19,20].

Evaluating the oxidative status of sperm is becoming increasingly important due to its strong correlation with male reproductive health [21]. This has led to the development of various methodologies to assess oxidative stress. Each method evaluates the activity of multiple antioxidant substances within the organic matrix. Although the results from these methods are not directly comparable, they are generally correlated [22].

Sperm freezing is crucial in stallions. Individual differences in sperm freezing sensitivity [23], along with agonist activities and selection choices, allow for reproductive activity even in individuals considered infertile [24].

This study aimed to analyze the freezing behavior of equine semen collected from several stallions. The semen was diluted with various extenders for transport and then with different commercial and semi-defined freezing extenders Antioxidant activity and nitric oxide content were assessed using methods that, in a previous study, effectively discriminated sperm functions in cooled donkey semen [25]. Analyzing the antioxidant capacity and RNS content of the extenders will provide new and unpublished information that we believe will contribute to a deeper understanding of this topic. This preliminary study will offer new insights into the different individual responses to freezing equine semen, laying the groundwork for subsequent investigations.

## 2. Materials and Methods

### 2.1. Materials

Unless otherwise indicated, all materials used for this study were purchased by Merck Life Science (Milan, Italy) and cell culture tested.

### 2.2. Animal Collection and Breeding Care

From April to May 2024, a single ejaculate was collected from ten stallions used as semen producers. These animals were of different breeds, mature (between 10 and 20 years old), and of proven fertility. All stallions were clinically healthy and were fed a standard diet consisting of mixed hay and basic concentrate, without antioxidant supplementation, with ad libitum access to water. They were housed in individual paddocks at the Regional Center of Equine Improvement (Centro Regionale di Incremento Ippico–S. Maria Capua Vetere, Caserta). This center is approved for equine semen collection by the Regional Government of Campania, Italy (authorization number: U 1500083 CE000642004), and operates under rigorous health and animal welfare protocols.

All animal procedures complied with the European Union guidelines (Directive 2010/63/EU and D. Lgs. 4/03/2014 n. 26) on the protection of animals used for scientific purposes, focusing on minimizing the number of animals used and reducing any pain or stress.

### 2.3. Sperm Collection, Dilution, and Shipping

Semen collection was carried out using a Missouri artificial vagina that was equipped with an in-line sterile gauze to remove the sperm gel fraction. Before sampling, each stallion underwent at least three preliminary semen collections scheduled twice weekly to empty the sperm epididymal reservoir. Each ejaculate volume was evaluated using a measuring cylinder, and then, after removing 2 mL of semen for plasma analyses, the remaining semen was divided into four aliquots, which were diluted 1: 2 (volume of semen: volume of extender) with three shipping and centrifugation extenders (SCEs). Each SCE was, later, associated with a proper freezing extender. One of these SCE (INRA 96^®^) was used for two freezing extenders, i.e., INRA-Freeze and HF-20. The SCEs were kept in sterile tubes at +4 °C and, before semen collection, they were heated to 37 °C. After semen addition, these tubes were inserted into a jar containing one liter of water at 37 °C placed inside a polystyrene box together with frozen cooling tiles for progressively decreasing sample temperature up to room temperature (RT = 20 °C) within approximately two hours from semen collection, ensuring a thermal decrease of approximately 0.10–015 °C/min. The three SCEs were:

No. 1. Equiplus^®^ (Minitüb GmbH, Tiefenbach, Germany) shows an undefined composition with caseinates, glucose, sucrose, lincomycin, and spectinomycin [26].

No. 2. BotuSemen^®^ (BotuSemen, Nidacon, Mölndal, Sweden) is a skim milk-based extender for chilled semen containing UHT sterilized skim milk powder, sugars, amino acids, antioxidants, and excipients, penicillin K 1g L^−1^ and gentamicin 1g L^−1^ [27].

No. 3. INRA 96^®^ (INRA 96, IMV Technologies, L’Aigle Cedex, France) containing HGLL, a solution of Hank’s salts, glucose and lactose with ultrapure water, buffered by 4.76 g·L^−1^ HEPES and supplemented with a purified fraction of caseins (native phosphocaseinate), 50,000 UI L^−1^ sodium penicillin, 50 mg L^−1^ gentamicin sulfate, and 0.25 mg L^−1^ amphotericin B [28].

### 2.4. Sperm Concentration and Kinetics

Upon arrival at the Laboratory of Biology and Technology of Animal Reproduction at the University of Basilicata, Potenza, all semen samples were assessed for sperm concentration and kinetics using a Makler chamber (Sefi-Medical Instruments, Haifa, Israel) and computer-assisted sperm analysis (SCA 5.0 system, Microptic, Barcelona, Spain). Following the evaluation of sperm concentration, each sample was diluted with the respective SCE to achieve a concentration of 30 × 10^6^ sperm mL^−1^ for SCA analysis. The samples were equilibrated for 2 min at 37 °C on a heated microscope stage. Spermatozoa with an average velocity of less than 10 μm s^−1^ were considered immotile. Sperm kinetics parameters assessed included the following: (i) the percentage of motile spermatozoa (TotMot); (ii) the percentage of progressive spermatozoa (Prog), defined as having an average path velocity greater than 30 μm s^−1^ and track straightness greater than 80%; (iii) curvilinear velocity (VCL, μm s^−1^); (iv) straight-line velocity (VSL, μm s^−1^); (v) average path velocity (VAP, μm s^−1^). For each sample, the tracks of at least one thousand spermatozoa were acquired and analyzed in duplicate by dividing them into two batches of more than 500 spermatozoa each.

### 2.5. Sperm Mitochondrial Membrane Potential (MMP) and Intracellular Hydrogen Peroxide (ROS) Content

For mitochondrial membrane potential (MMP), sperm samples were treated with JC-1, a fluorescent dye exhibiting potential-dependent accumulation in mitochondria. Following excitation at 488 nm wavelength, it emits two peaks at ~595 nm (F_0_B) and ~535 nm (F_0_A) wavelengths. The fluorescence intensity ratio of these fluorescent peaks expresses the mitochondrial membrane potential (F_0_B/F_0_A = MMP) [29,30]. An estimated number of 1 × 10^6^ spermatozoa for each sperm sample was washed with PBS-PVA by centrifugation (300× *g* for 5 min), resuspended in 200 μL PBS-PVA, and incubated with 1.5 μM JC-1 in the dark at RT for 30 min. After incubation, sperm suspensions were centrifuged at 300 g for 5 min, the supernatant removed and the pellet resuspended in 800 μL PBS-PVA and incubated for 30 min. Each sample was placed in a quartz microtube and read using a spectrofluorometer (Cary Eclipse, Agilent Technologies, Rome, Italy). Fluorescent spectra from 500 to 620 nm were recorded in duplicate samples.

Intracellular hydrogen peroxide (H_2_O_2_) content (ROS) was assessed with the 2′,7′-dichlorodihydrofluorescein diacetate (H_2_DCFDA) [19]. After a double wash with PBS-PVA by centrifugation (300× g for 5 min), sperm aliquots of 2 x 10^6^ sperm cells mL^−1^ were stained with 10 μM H_2_DCFDA for 30 min in the dark at RT, washed and further incubated for 30 min in 800 μL PBS-PVA. After incubation, samples were analyzed with a spectrofluorometer, as above. The peak fluorescence intensity at ~525 nm following an excitation wavelength of 488 nm is proportional to the intracellular H_2_O_2_ levels and measured as arbitrary units (a.u.).

### 2.6. Sperm Freezing

Simultaneously with the evaluation of sperm properties, four aliquots from each ejaculate, diluted with the three SCEs, were centrifuged at 300× *g* for 10 min. After collecting and storing the conditioned extenders at −80 °C, each sperm pellet was resuspended with one of four freezing extenders to achieve a sperm concentration of 100 × 10^6^ sperm mL^−1^. These samples were then incubated for specific times and temperatures (see below) to allow diffusion and equilibration between the extracellular and intracellular compartments of the CPAs. The equilibration time was determined according to literature recommendations or manufacturer guidelines, as indicated below. These four freezing extenders, like the SCEs mentioned above, are products successfully used for stallion sperm freezing. Three are commercial products with proprietary compositions, while HF-20 is a semi-defined medium produced in our laboratory with the composition listed below.

No. 1. Spectrum Duo Red^®^: Equiplus was replaced with Spectrum Duo Red^®^, which contains buffers, sugars, salts, antibiotics (0.5 g L^−1^ amikacin sulfate and 1 × 10^6^ IU L^−1^ penicillin G potassium), 20% egg yolk (EY), and a mix of CPAs. Equilibration time = 5 min at RT. After equilibration, the semen doses were placed at −20 °C for 3.5 min and then exposed to liquid nitrogen (LN_2_) vapors.

No. 2. BotuCrio^®^: BotuSemen was replaced with BotuCrio^®^, which contains sugars, antioxidants, amino acids, 7.3% EY, and excipients. It is also supplemented with 1% glycerol, 4% methylformamide, and 0.5 g/L gentamicin [31]. The glucose concentration is 99.0 ± 6.1 mM. Equilibration time = 20–30 min at +4 °C. Semen doses were placed in a refrigerator at +4 °C for 20–30 min and then exposed to LN_2_ vapors.

No. 3. INRA Freeze^®^: One of the two samples diluted with INRA 96 was replaced with INRA Freeze^®^, which consists of INRA 96 supplemented with 2.5% glycerol and EY [32,33]. Equilibration time = 90 min with a decreasing temperature gradient down to +4 °C [34]. The tubes with semen were initially dipped in a bucket of water at +20 °C and then placed in a refrigerator set at +4 °C for 60 min. Afterward, these tubes were transferred to a cold handling cabinet at +4 °C, and the sperm doses were packaged in straws. The sperm doses were then placed in a +4 °C refrigerator for an additional 15 min before exposure to LN_2_ vapors.

No. 4. HF-20: The other of the two samples diluted with INRA 96 was replaced with HF-20, which contains 5 g glucose, 0.3 g lactose, 0.3 g raffinose, 0.15 g sodium citrate, 0.05 g sodium phosphate, 0.05 g potassium sodium tartrate, 10% EY, 25,000 IU penicillin, 0.08 g streptomycin, 3% glycerol, and ultrapure water up to 100 mL [35]. The fresh EY used in both HF-20 and Spectrum Duo Red was pooled to reduce biological variability. Equilibration time = 90 min with a decreasing temperature gradient down to +4 °C [35]. The procedure for packaging and freezing the semen doses was the same as that described above for INRA Freeze^®^.

### 2.7. Sperm Packaging, Freezing, and Thawing

Semen doses were packaged with individually labeled 0.5 mL polyvinyl chloride straws (IMV Technologies, L’Aigle, France) that had previously been placed at +4 °C for use with sperm freezing extenders (nos. 2, 3, and 4). After loading with semen, all straws were sealed with a filling powder (Poudre de bouchage bleue, IMV Technologies) and placed horizontally into the freezing rack (distribution block for 40 medium straws, IMV Technologies). At the time of freezing, the rack was set in a polystyrene box containing LN_2_, with the straws 4 cm above LN_2_ for 10 min, and then plunged into LN_2_ for storage. After 1 week from freezing, four straws for each extender of each stallion were thawed in a water bath at 37 °C for 30 s immediately before semen analyses. After the first centrifugation to remove the freezing conditioned medium that was collected and stored at −80 °C for biochemical analyses, the pellet of each sample was resuspended in 400 µL INRA96, placed on a discontinuous Percoll gradient (70/40) [36], and centrifuged for 15 min at 1000× *g* to remove EY residues that would have disturbed subsequent sperm analyses. The pellet obtained was resuspended in all the samples with INRA 96 and evaluated under the microscope for motility and kinetics. The remaining sperm suspension was washed with PBS-PVA and used for assessing sperm mitochondrial activity, and H_2_O_2_ content.

### 2.8. pH and Osmolarity Evaluation

The pH and osmolality of all the semen extenders were measured using a Jenway 3020 pH meter (Jenway, London, UK) and micro-osmometer (Digital Osmometer, Roebling, Berlin, Germany).

### 2.9. Biochemical Analyses

The total antioxidant capacity (TAC) was assessed by ABTS and ferric reducing antioxidant power (FRAP) assays in semen extenders, semen plasma, and conditioned extenders before and after sperm freezing, on three different batches of each sample. Moreover, an estimate of the reactive nitrogen species (RNS) content of the samples was conducted with nitric oxide (NO) stable metabolites (NOx) content assay. Upon the semen’s arrival in the laboratory and after thawing, all sperm samples were subjected to first centrifugation (300× *g* for 10 min) to separate the spermatozoa from the conditioned extender. Further centrifugation (2500× *g* for 10 min at +4 °C) was carried out on the supernatant to eliminate any cellular component and allow biochemical analyses of the samples.

Total antioxidant activity, based on the reduction of colored 2,2′-azinobis-(3-ethylbenzothiazoline-6-sulfonic acid) radical cation (ABTS^·+^), was measured according to Erel [37]. Briefly, 5 μL of the samples were mixed with 200 μL of acetic acid–sodium acetate buffer (pH 5.8) and measured at 660 nm with a microplate reader (Bio-Rad, mod. 550, Segrate, Milan, Italy). Thereafter, 20 μL of acetic acid–sodium acetate buffer (pH = 3.6) containing 2 mM H_2_O_2_ and 10 mM ABTS were added to these mixtures. After 5 min of incubation at RT, the optical densities (ODs) were read again at 660 nm. The assay was calibrated with ascorbic acid (AA) and the results are presented as AA equivalents (μM).

FRAP assay was carried out following the original method of Benzie and Strain [38]. In brief, sodium acetate buffer (pH 3.6), tris(2-pyridyl)-s-triazine, and iron(III) chloride hexahydrate were mixed to generate a FRAP fresh prepared solution. Later, 10 μL of samples in duplicates were added to 300 μL FRAP solution and, after 5-min incubation at 37 °C, the ODs of the reaction mixture were read at 600 nm with the microplate reader. The assay was calibrated with iron sulfate heptahydrate (FeSO_4_·7H_2_O) and the results were reported as FeSO_4_ ·7H_2_O equivalents (μM).

Nitric oxide stable metabolites (NOx), which include the sum of nitrite (NO_2_^−^) and nitrate (NO_3_^−^), were measured using the Griess reagent, following the protocol described by Miranda et al. [39]. Briefly, 100 µL of deproteinized samples were treated with 100 µL of 0.8% vanadium (III) chloride in 1 M HCl to reduce nitrate to nitrite. Subsequently, 100 μL of Griess reagent (comprising 0.2% sulfanilamide and 0.1% N-1-naphthyl)ethylenediamine) was added. After 30-min incubation at 30 °C, ODs were read at 540 nm using the microplate reader with the assay calibrated against sodium nitrate (NaNO_3_). Results were expressed as NaNO_3_ equivalents (μM).

### 2.10. Statistical Analysis

Data were analyzed by two-way ANOVA (independent variables: stallion and extender) by using the open-source software JASP 0.18.3 (University of Amsterdam, Amsterdam, NL). Multivariate linear regression analysis compared sperm functions as well as antioxidant capacity and NOx values in refrigerated and frozen/thawed samples; this model included stallion and extender effects. Before the analyses, percentage values were transformed in arcsine, whereas for pH_i_, H^+^ concentrations were log-transformed. Normal data distribution and homogeneity of variance were assessed by the Shapiro–Wilks test and Levene’s test, respectively. Pair-wise comparisons of the means were performed with the Turkey test. The threshold of *p* < 0.05 was used as the minimum level of statistical significance. Data are shown as the mean ± standard deviation (SD).

## 3. Results

Significant analytical differences were observed between the blank extenders for both fresh and frozen semen, as shown in Table 1. Among the SCEs, BotuSemen exhibited the highest osmolarity and ABTS-based antioxidant activity, as well as the lowest pH. In the group of freezing extenders, BotuCrio demonstrated the highest osmolarity and was the only extender with ABTS-based antioxidant activity. Additionally, Spectrum Duo Red had a higher osmolarity compared to INRA Freeze and HF-20 and shared similar FRAP-based antioxidant activity with BotuCrio.

Analysis of seminal plasma to assess antioxidant activities and NOx content (Table 2) revealed significant variability among stallions (*p* < 0.01) for both antioxidant activity estimates (ABTS and FRAP) and NOx levels.

By correlating the plasma values of ABTS, FRAP, and NOx with sperm kinematic parameters, MMP, and ROS in both pre- and post-freezing semen, we obtained some interesting insights. In refrigerated semen, there were no significant correlations between ABTS values and total or progressive sperm motility, MMP, or ROS. However, significant correlations were observed with curvilinear velocity (VCL, R = 0.646, *p* = 0.043), straight-line velocity (VSL, R = 0.835, *p* = 0.003), and average path velocity (VAP, R = 0.840, *p* = 0.002). For FRAP-based antioxidant activity, no statistically significant correlations were found with any sperm kinematic parameters, MMP, or ROS. NOx levels in seminal plasma were significantly correlated with VCL (R = 0.763, *p* = 0.010) and MMP (R = 0.677, *p* = 0.032). However, no statistically significant correlations were observed between these seminal plasma parameters and post-thaw sperm functions.

In conditioned SCEs (cSCEs), FRAP-based TAC was detected in all samples, while ABTS-based TAC was detected in the majority (27 out of 30; 90%) and NOx was found in only 7 out of 30 (23%) samples (Appendix A). A comparison of results among the three cSCEs (Table 3) showed no significant differences, although their values were, on average, higher than those in the blank extenders. In conditioned freezing extenders, FRAP-based TAC was detected in all samples, ABTS-based TAC was found in only 14 of 40 (35%) samples, and NOx in 16 of 40 (40%) samples (Appendix A). Unlike the cSCEs, significant differences were observed among the freezing extenders in both TAC assays and NOx values. Notably, the highest ABTS-based TAC levels were detected in conditioned INRA Freeze, even though the blank extender did not exhibit this activity. Furthermore, no correlation was found between ABTS- and FRAP-based antioxidant activities and NOx values in conditioned extenders compared to either blank extenders or stallion seminal plasma except for NOx whose values in cSCEs were related to those found in the stallion seminal plasma (R = 0.796; *p* = 0.006).

Appendix A present the data (mean ± SD) for individual stallions regarding sperm kinetics, MMP, and ROS, as well as the antioxidant activity (ABTS and FRAP) and NOx values of the conditioned extenders before and after freezing. In cooled sperm, significant individual variations (*p* < 0.01) were found across all variables related to sperm kinetics, MMP, and ROS. However, no significant differences were observed among the SCEs (Table 4). Similarly, in frozen/thawed spermatozoa, significant differences (*p* < 0.01) were detected among individual stallions for all parameters related to sperm kinetics, MMP, and ROS. ANOVA analysis revealed that, apart from a small significant difference (*p* < 0.05) in average path velocity (VAP) values, no significant variations were observed among the freezing extenders concerning sperm kinetics, MMP, and ROS (Table 4). A significant interaction (*p* < 0.01) was also found between stallions and extenders for all evaluated variables.

Figure 1 presents regression lines comparing the results of sperm kinetics, MMP, and ROS, as well as antioxidant activity based on both ABTS and FRAP methods, and NOx values in conditioned extenders before and after freezing. This analysis considers the effects of both stallions and extenders in the model. The type of semen storage (cooled vs. frozen) had a significant impact on all the analyzed parameters. Stallions had a significant effect on TotMot, VCL, VSL, and VAP. Extenders had a significant effect on VSL, VAP, NOx levels, and FRAP-based antioxidant activity. Moreover, a significant correlation was found between antioxidant activities measured by the FRAP and ABTS assays (R = 0.461; *p* < 0.01) in conditioned extenders from cooled and frozen samples. In contrast, no significant correlation was found between the ABTS and FRAP assays when performed in blank extenders (R = 0.162; *p* = 0.614) or stallion seminal plasma (R = 0.297; *p* = 0.405).

## 4. Discussion

The freezing sensitivity of stallion semen is largely attributable to individual variation [40]; however, this study highlights how environmental factors, such as the freezing extender, may affect sperm functional outcomes by changing the environmental antioxidant activity and nitric oxide content.

To investigate the mechanisms underlying individual differences in sperm freezing sensitivity, the antioxidant capacity and nitric oxide stable metabolites (NOx) were measured in the seminal plasma and both blank and conditioned extenders.

ABTS-based total antioxidant capacity (TAC) has been used in equine semen to evaluate the effect of dietary antioxidant supplementation on the quality of refrigerated [41] or frozen [42] semen. The FRAP-based assay for antioxidant activity has been less commonly used for equine semen, though it has been evaluated in seminal plasma and correlated with some post-freezing sperm functions [43]. The role of nitric oxide (NO) in sperm fertility is well documented [44]. Like reactive oxygen species (ROS), small quantities of NO play crucial roles in many physiological mechanisms [45]; however, high concentrations are associated with pathological conditions [46]. 

In our study, the antioxidant activity and NOx content revealed substantial differences among extenders, along with variations in osmolarity and pH. Such variations were larger among freezing extenders than SCEs, likely due to differences in the CPA content of commercial freezing extenders, whose exact compositions are often not disclosed. The first two extenders (Spectrum Duo Red and BotuCrio) showed physicochemical characteristics, antioxidant activity, and RNS content that significantly differed from the other two extenders, which were united by the same freezing procedure. However, no statistically significant differences were found among extenders regarding sperm functionality. 

The antioxidant potential and NOx content were also assessed in the seminal plasma to identify possible relationships between these parameters and sperm function. Significant correlations were found between ABTS-based TAC and some sperm kinematic parameters, whereas NOx values were significantly correlated with VAP and MMP. These results agreed with those found in cooled donkey semen [30], although, in donkeys, ABTS- and FRAP-based TAC and NOx values were well correlated with numerous sperm function parameters. On the other hand, no correlation was detected between these seminal plasma components and post-freezing sperm function. These findings align with those of a recent study evaluating the effect of enzymatic and non-enzymatic antioxidant activities in horse seminal plasma on post-freezing sperm functions [43]. Catalán et al. did not find statistically significant differences between good and poor stallion sperm freezers based on the FRAP assay. However, other assays evaluating non-enzymatic antioxidant activity, such as Trolox equivalent antioxidant capacity (TEAC), and enzymatic antioxidant activity, such as paraoxonase type 1 (PON1) and superoxide dismutase (SOD), demonstrated a good relationship with stallion sperm cryotolerance. 

Another source of information on the role of antioxidant activity and NOx in sperm freezing was investigated in the conditioned extenders. In the conditioned shipping and centrifugation extenders (cSCEs), the average values of the two TAC assays were higher than those of the blank extenders. This could be partly attributed to the seminal plasma component, which constitutes approximately one-fifth of the cSCE. The comparison among cSCEs regarding sperm functions did not show significant differences, whereas the antioxidant activities and NOx contents of the various freezing-conditioned extenders were significantly different. In particular, the INRA freeze showed the highest ABTS activity, although this activity was not found in the blank extender. We do not have an explanation for this finding or the source of this antioxidant activity. This conditioned-freezing extender also showed the lowest FRAP activity, which agrees with the blank extender results. To our knowledge, a residual evaluation of the antioxidant activity in the extender at the end of its incubation with spermatozoa has never been previously assessed. These assessments highlighted significant differences between extenders, although they were unable to identify substantial changes in sperm function. 

Comparing extenders provides interesting insights. The lack of significant differences among SCEs, combined with the low chemical–physical differences in TAC and NOx, suggests that the choice of extender may be irrelevant, at least considering the short time between semen collection and freezing. Evidence has shown that collected stallion semen, when maintained at refrigeration temperatures, undergoes progressive alterations in sperm morphology and functionality, rendering the sample unusable and infertile after 72 h of conservation at 4–5 °C. Better performance seems to derive from sperm storage at moderate temperatures (15–22 °C) (for review see [47]). These are the conditions to which we submitted our seminal samples before applying freezing procedures, allowing us to work more easily at room temperature. In addition, a slow thermal decrease, as adopted in this study, appears to improve stallion sperm performance [48]. 

Different speculations can be made regarding freezing extenders. Although freezing extenders did not show significant differences in maintaining post-thawing sperm functions, they significantly differed in antioxidant activity and NOx values. The lack of statistically significant differences between extenders may be due to the high variability of results linked to strong individual responses and extender-stallion interactions. Each individual showed different sperm functionality with each extender, which in some cases enhanced and in others hindered sperm cryotolerance. Overall, these results showed high variability, making it impossible to identify an ideal freezing extender for all stallions. Based on this finding, future research may aim to verify whether this individual behavior is consistent in repeated samples from the same individual or occurs randomly. 

Differences between extenders have been widely described in the literature, with mixed results. For example, Ramires Neto and colleagues [49] found higher post-freezing sperm functionality using BotuCrio compared to INRA Freeze and EquiPRO CyoGuard. However, differences between BotuCrio and INRA Freeze diminished when focusing on good freezer stallions. Similar results were reported by Gutiérrez-Cepeda et al. [50], who obtained better post-freezing sperm function with BotuCrio compared to INRA Freeze and Lac-Edta. Neuhauser et al. [51], comparing four freezing extenders (INRA Freeze, BotuCrio, Equiplus Freeze, and Gent Freeze), found no differences in sperm function between INRA Freeze and BotuCrio while recording significantly lower results with the other two freezing extenders. Other researchers [52] found no differences in sperm functions post-freezing using HF-20, INRA Freeze, or Equiplus Freeze. Although stallion sperm demonstrated an extender-dependent individual response that contributed to modifying the individual’s sensitivity to sperm freezing, a universally superior extender for all stallions has yet to be identified. Private companies are currently developing new products to achieve this goal, although detailed information in this regard is often lacking. In the present study, different commercial extenders were compared with a semi-defined extender (HF-20), which contains known elements and an undefined component such as egg yolk (EY) [35,52]. Although simple and unsophisticated, this extender produced similar results compared to other commercial extenders and met the necessary requirements for future studies. 

Focusing on freezing extenders involves considering the entire freezing procedure, which is characterized by different equilibrium times and cooling rates. Additionally, substantial changes have been made to the type and concentration of CPAs, such as replacing glycerol with better-tolerated amides [12], altering the EY or EY replacement quota [28,53], and replacing skimmed milk with milk proteins [54]. In our study, we did not detect significant differences in sperm function between the freezing extenders used; however, by highlighting different antioxidant activities and NOx, we have identified discriminating elements that could be useful for subsequent evaluations. 

## 5. Conclusions

A comparison of different stallion semen transport and freezing extenders has revealed a significant stallion–extender interaction and a strong individual tolerance to freezing. By assessing antioxidant activity using two different assays and measuring NOx, we found considerable stallion-dependent variability in the seminal plasma, as well as in both blank and conditioned extenders. However, these variations were not associated with differences in sperm function among the extenders. Customizing the freezing procedures and extenders appears to be an effective strategy to optimize individual tolerance to semen freezing and improve post-freezing sperm function.

## Figures and Tables

**Figure 1 animals-14-02465-f001:**
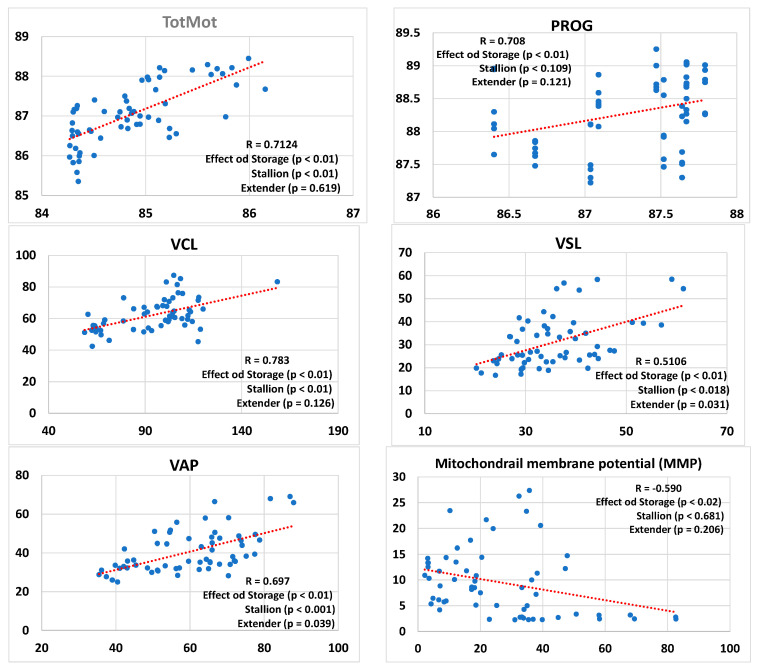
Regression lines and correlation coefficients (R) between refrigerated and frozen/thawed (storage) sperm including the effects of the individual stallions and the extenders for sperm kinetics–sperm total (TotMot) and progressive (Prog) motility, curvilinear velocity (VCL), straight-line velocity (VSL), average path velocity (VAP)–mitochondrial membrane potential (MMP), and hydrogen peroxide content (ROS) as well as antioxidant activity (ABTS and FRAP) and nitric oxide stable metabolites (NOx) assessed in the conditioned extenders. The percentages of total (TotMot) and progressive (Prog) sperm motility have been expressed in arcsine.

**Table 1 animals-14-02465-t001:** Osmolarity, pH, ABTS- and FRAP-based total antioxidant activity, and nitric oxide stable metabolites (NOx) measured in the shipping/centrifugation (SCE) and freezing extenders.

Shipping/Centrifugation Extenders (SCEs)	Osmolarity	pH	ABTS	FRAP	NOx
	mOsm L^−1^		µM	µM	µM
Equiplus	324 ± 3 ^A^	6.85 ± 0.05	61.1 ± 2.5 ^A^	163 ± 7 ^A^	n.d.
BotuSemen	390 ± 5 ^B^	6.78 ± 0.03 ^a^	212.3 ± 4.9 ^BC^	187 ± 10 ^A^	n.d.
INRA96	317 ± 1 ^A^	6.91 ± 0.07 ^b^	32.0 ± 7.6 ^BD^	236 ± 7 ^B^	n.d.
Freezing Extenders					
Spectrum Duo Red	985 ± 5 ^A^	6.66 ± 0.11 ^A^	n.d.	1234 ± 10 ^A^	52.7 ± 2.1 ^A^
BotuCrio	1321 ± 7 ^BC^	6.73 ± 0.13 ^B^	79.1 ± 4.0	1213 ± 15 ^A^	52.3 ± 3.2 ^A^
INRAFreeze	734 ± 3 ^BDE^	6.85 ± 0.07 ^BC^	n.d.	379 ± 5 ^B^	65.3 ± 3.2 ^B^
HF-20	809 ± 3 ^BDF^	7.02 ± 0.05 ^BD^	n.d.	360 ± 5 ^B^	n.d.

Different letters in the same column indicate significant differences (A, B, and C, D, and E, F; *p* < 0.01) (a, b; *p* < 0.05); n.d.: not detectable.

**Table 2 animals-14-02465-t002:** ABTS- and FRAP-based antioxidant activity and nitric oxide stable metabolites (NOx) detected in the seminal plasma of the stallion ejaculates.

	ABTS	FRAP	NOx
Stallion	µM	µM	µM
No. 1	129 ± 4	393 ± 7	n.d.
No. 2	245 ± 6	550 ± 6	n.d.
No. 3	2202 ± 11	405 ± 6	n.d.
No. 4	28 ± 4	556 ± 15	64 ± 4
No. 5	910 ± 11	934 ± 18	64 ± 6
No. 6	780 ± 4	538 ± 10	106 ± 6
No. 7	305 ± 8	259 ± 6	n.d.
No. 8	673 ± 11	489 ± 8	350 ± 13
No. 9	1611 ± 29	972 ± 20	529 ± 11
No. 10	708 ± 10	301 ± 6	54 ± 5

n.d.: not detectable.

**Table 3 animals-14-02465-t003:** ABTS- and FRAP-based antioxidant activity and nitric oxide stable metabolites (NOx) (mean ± SD) detected in the conditioned shipping/centrifugation and freezing extenders.

Conditioned Shipping/	ABTS	FRAP	NOx
Centrifugation Extenders (cSCEs)	µM	µM	µM
Equiplus	353 ± 301	356 ± 134	20 ± 40
BotuSemen	432 ± 344	347 ± 167	9 ± 29
INRA 96	443 ± 428	415 ± 126	36 ± 91
**Conditioned Freezing Extenders**			
Spectrum Duo Red	22 ± 46 ^A^	930 ± 152 ^A^	96 ± 154 ^A^
BotuCrio	88 ± 167 ^A^	728 ± 215 ^BC^	112 ± 99 ^A^
INRA Freeze	659 ± 365 ^B^	388 ± 98 ^BD^	4 ± 12 ^B^
HF-20	n.d.	704 ± 111 ^BC^	n.d.

Different letters in the same column indicate significant differences (A, B, and C, D; *p* < 0.01); n.d.: not detectable.

**Table 4 animals-14-02465-t004:** Sperm kinematic parameters, mitochondrial membrane potential (MMP), and hydrogen peroxide content (ROS) as assessed in shipping/centrifugation extenders (SCEs) and freezing extenders.

	Shipping/Centrifugation Extenders (SCEs)			Freezing Extenders			
	Equiplus	BotuSemen	INRA 96	Spectrum Duo Red	BotuCrio	INRA Freeze	HF-20
TotMot (%)	78.2 ± 14.4	80.0 ± 16.3	84.3 ± 16.2	28.4 ± 14.8	25.7 ± 13.7	30.5 ± 13.1	25.6 ± 14.9
Prog (%)	23.1 ± 7.0	19.6 ± 7.8	24.5 ± 5.5	11.7 ± 6.6	9.6 ± 6.2	8.6 ± 5.5	7.5 ± 5.1
VCL (µm/s)	95.7 ± 23.3	93.1 ± 20.1	94.1 ± 18.3	65.5 ± 10.9	62.8 ± 10.6	58.5 ± 7.9	59.4 ± 9.5
VSL (µm/s)	35.8 ± 10.2	34.9 ± 9.3	36.1 ± 8.1	34.4 ± 12.1	32.2 ± 11.9	26.4 ± 7.4	27.1 ± 7.7
VAP (µm/s)	59.3 ± 14.8	60.3 ±13.7	59.8 ±12.3	44.7 ± 12.3 ^a^	41.0 ± 11.3	36.1 ± 7.3 ^b^	36.8 ± 8.1
MMP (F_0_B/F_0_A)	34.9 ± 13.6	25.6 ± 20.2	24.8 ± 21.3	8.9 ± 6.8	10.1 ± 7.0	9.5 ± 6.4	9.2 ± 5.6
ROS (a.u.)	24.3 ± 4.3	24.1 ± 4.3	23.3 ± 2.8	12.7 ± 5.6	13.3 ± 6.9	13.1 ± 6.6	13.6 ± 8.8

TotMot (total sperm motility), Prog (progressive sperm motility), VCL (curvilinear velocity), VSL (straight-line velocity), VAP (average path velocity). Different letters in the same row indicate significant differences (a, b; *p* < 0.05).

## Data Availability

No new data were created or analyzed in this study. Data sharing is not applicable to this article.

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
