# Peer review of "Stallion Sperm Freezing with Different Extenders: Role of Antioxidant Activity and Nitric Oxide Production"

_animals, 2024, doi:10.3390/ani14172465_

Round 1

Reviewer 1 Report

Comments and Suggestions for Authors

This is an interesting study investigating the effects on sperm kinetics, mitochondrial membrane potential and oxidative stress of three different extenders used for freezing equine semen.

The topic is certainly interesting because, as reported by the authors, in equids there is a wide subjective variability in the effects of semen cryopreservation (bad and good freezers stallions).

The results obtained can provide a valuable contribution in the field of equine semen freezing, as they demonstrate how important it is to adapt the freezing protocol to the semen material.

The work is overall very well structured: the materials and methods are described exhaustively and the results obtained are well represented by tables and graphs.

The discussions are also well structured.

I just have a few small suggestions to improve the overall quality of the manuscript before publication.

The introduction provides a good basic background, particularly regarding the freezing process and the risks of damaging sperm. however, I believe that this part can be slightly summarized, to give greater emphasis to other topics, central to the study, such as membrane potential and oxidative stress.

Furthermore, some concepts reported in the discussions (lines 360-383) could be summarized and/or moved to the introduction.

Finally, the units are not shown in the tables presented as supplementary material. I suggest placing them in column headings or as footnotes.

Author Response

Reviewer 1

This is an interesting study investigating the effects on sperm kinetics, mitochondrial membrane potential and oxidative stress of three different extenders used for freezing equine semen.

The topic is certainly interesting because, as reported by the authors, in equids there is a wide subjective variability in the effects of semen cryopreservation (bad and good freezers stallions).

The results obtained can provide a valuable contribution in the field of equine semen freezing, as they demonstrate how important it is to adapt the freezing protocol to the semen material.

The work is overall very well structured: the materials and methods are described exhaustively and the results obtained are well represented by tables and graphs.

The discussions are also well structured.

  1. We sincerely thank the Reviewer for their efforts in improving our paper and for their positive feedback.

I just have a few small suggestions to improve the overall quality of the manuscript before publication.

The introduction provides a good basic background, particularly regarding the freezing process and the risks of damaging sperm. however, I believe that this part can be slightly summarized, to give greater emphasis to other topics, central to the study, such as membrane potential and oxidative stress.

  1. The introduction is designed to summarize the key issues related to semen freezing and to introduce the topic of oxidative stress, particularly focusing on the potential role of CPAs in this process. We greatly appreciate the Reviewer’s valuable suggestion. In response, we have revised the text by adding several sentences, some of which were moved from the discussion, to highlight the significance of oxidative stress and antioxidant activities, while also introducing the role of mitochondria (L 87-95).

Furthermore, some concepts reported in the discussions (lines 360-383) could be summarized and/or moved to the introduction.

  1. We agree with the Reviewer’s suggestion and have summarized this section, relocating it to the introduction as advised.

Finally, the units are not shown in the tables presented as supplementary material. I suggest placing them in column headings or as footnotes.

  1. We are grateful for the suggestion, which we have promptly implemented.

Reviewer 2 Report

Comments and Suggestions for Authors

In this maniscript the author present data aimed to show the extender effect upon stallin smene freezing. These results are relevant interms of basic science and in dustry. I have onle some minors question /suggestions

Line 38-39 I do not see this point. Which are the "mechanisms" since this is a rather descriptive paper were the author shows only correlations but not caisative evidences of any studies parameter

Line 265-269 please give a more extense description of the procedure to evaluate NOx, remember that other authors migth want to reproduce your results and this is highly dependendable upon the quality of the procedure description.

Author Response

Reviewer 2

In this maniscript the author present data aimed to show the extender effect upon stallin smene freezing. These results are relevant interms of basic science and industry. I have onle some minors question /suggestions

  1. We sincerely thank the Reviewer for their efforts in improving our paper and for their positive feedback.

Line 38-39 I do not see this point. Which are the "mechanisms" since this is a rather descriptive paper were the author shows only correlations but not caisative evidences of any studies parameter

  1. We agree that our enthusiasm may have led us to overstate our case. In reality, this work serves as a precursor to future studies and is intended only as a necessary foundation. Consequently, we have revised the sentence to be more accurate (L 38-39).

 Line 265-269 please give a more extense description of the procedure to evaluate NOx, remember that other authors migth want to reproduce your results and this is highly dependendable upon the quality of the procedure description.

  1. Thank you for your insightful observation. We also recognized that our description of the NOx assay methodology was too vague. The effort to avoid plagiarism can sometimes result in such oversights. We have now provided a clearer and concise summary of the methodology used (L 292-299). 
